# Policy options for surgical mentoring: Lessons from Zambia based on stakeholder consultation and systems science

Henk Broekhuizen[1,2]*, Martilord Ifeanyichi[1,3], Mweene Cheelo[4], Grace Drury[5], Chiara Pittalis[6], Etiënne Rouwette[7], Michael Mbambiko[4], John Kachimba[4], Ruairí Brugha[8], Jakub Gajewski[6], Leon Bijlmakers[1]

1 Dept. Health Evidence, Radboud University Medical Center, Nijmegen, The Netherlands, 2 Dept. Health and Society, Wageningen University and Research, Wageningen, The Netherlands, 3 EMAI Health Systems and Health Services Consulting, Nijmegen, The Netherlands, 4 Department of Surgery, Surgical Society of Zambia, University Teaching Hospital, Lusaka, Zambia, 5 Nuffield Department of Orthopaedics, Rheumatology and Musculoskeletal Sciences, University of Oxford, Oxford, United Kingdom, 6 Institute of Global Surgery, Royal College of Surgeons in Ireland, Dublin, Ireland, 7 Institute for Management Research, Radboud University, Nijmegen, The Netherlands, 8 Department of Epidemiology & Public Health, Royal College of Surgeons in Ireland Division of Population Health Sciences, Dublin, Ireland

* henk.broekhuizen@wur.nl

**Data Availability Statement:** All relevant data are within the manuscript and its Supporting Information files.

## Abstract

### Background

Supervision by surgical specialists is beneficial because they can impart skills to district hospital-level surgical teams. The SURG-Africa project in Zambia comprises a mentoring trial in selected districts, involving two provincial-level mentoring teams. The aim of this paper is to explore policy options for embedding such surgical mentoring in existing policy structures through a participatory modeling approach.

### Methods

Four group model building workshops were held, two each in district and central hospitals. Participants worked in a variety of institutions and had clinical and/or administrative backgrounds. Two independent reviewers compared the causal loop diagrams (CLDs) that resulted from these workshops in a pairwise fashion to construct an integrated CLD. Graph theory was used to analyze the integrated CLD, and dynamic system behavior was explored using the Method to Analyse Relations between Variables using Enriched Loops (MARVEL) method.

### Results

The establishment of a provincial mentoring faculty, in collaboration with key stakeholders, would be a necessary step to coordinate and sustain surgical mentoring and to monitor district-level surgical performance. Quarterly surgical mentoring reviews at the provincial level are recommended to evaluate and, if needed, adapt mentoring. District hospital administrators need to closely monitor mentee motivation.

**Funding:** The SURG-Africa study is funded by the European Union's Horizon 2020 Programme for Research and Innovation, under grant agreement no: 733391. The funders had no role in study design, data collection and analysis, decision to publish, or preparation of the manuscript.

**Competing interests:** The authors have declared that no competing interests exist.

**Abbreviations:** SURG-Africa, acronym for the 'scaling up safe surgery for district and rural populations in Africa' project; CLD, causal loop diagram; MARVEL, method to analyse relations between variables using enriched loops; NPC, non-physician clinician; ML, medical licentiate; NSOAP, national surgical, obstetric, and anesthesia strategic plan; MOH, ministry of health; COST-Africa, acronym for the 'clinical officer surgical training in Africa' project; GMB, group model building; LCH, Livingstone central hospital; UTH, university teaching hospital; DH, district hospital; HC, health center; CH, central hospital; HRM, human resource management; OT, operating theatre; CMS, central medical stores; PMF, provincial mentoring faculty.

## Conclusions

Surgical mentoring can play a key role in scaling up district-level surgery but its implementation is complex and requires designated provincial level coordination and regular contact with relevant stakeholders.

## Background

Access to surgery in Zambia remains low, especially in rural areas where about two thirds of the population reside [1]. The district hospitals that serve these areas experience severe manpower shortages [2, 3]. To address these shortages, Zambia has invested in the surgical training of a non-physician clinician (NPC) cadre called medical licentiates (MLs) [4]. The costs of training and retaining this cadre are lower than for medical doctors, with higher retention rates in rural areas [5]. Earlier studies in Zambia and elsewhere have shown that MLs deliver basic surgical care, with patient outcomes similar to surgery performed by specialists [6, 7]. Nevertheless, NPCs at district hospitals face a variety of professional, regulatory, and infrastructural challenges [5, 8]. Because of their proximity to rural dwellers, the Zambian national surgical, obstetric, and anesthesia strategic plan (NSOAP) emphasizes strengthening of surgically trained medical doctors and MLs at district hospitals [9].

Mentoring and coaching–terms that are often used interchangeably–are frequently recommended as effective ways of not only strengthening the clinical management of various types of illnesses by professionals practicing in peripheral hospitals, but also the managerial performance of the same institutions [10, 11]. As such they are considered a core component of health systems strengthening to achieve universal health coverage. In 2012, the Ministry of Health (MOH) in Zambia started a national multi-disciplinary mentorship program. Its overall goal is to improve the clinical environment in health facilities, as a complement to the strategy of supportive supervision, the goal of which is to audit or monitor the quality of care. The Ministry developed a mentorship training package [12] and in 2017 its Clinical Care and Diagnostic Services department issued a new version of the Mentorship guidelines for health workers [13]. The guidelines aim to support national, zonal and provincial mentorship teams composed of professionals with substantial expertise in four clinical disciplines: surgery, internal medicine, pediatrics & child health, and obstetrics & gynecology. The mentors are expected to build relationships with teams of mentees at lower levels of the health system, in particular at district hospitals, so as to strengthen district-level capacity. Zambia's current five-years National Health Strategic Plan does mention 'mentorship and supportive supervision' as one of several strategies to enhance quality in training and health service delivery; however, it lacks elaboration and is not included as a budget line in the national health budget [14].

After an earlier positive pilot in the COST-Africa project [15], the SURG-Africa trial is currently evaluating the effectiveness and sustainability of surgical mentoring in three countries: Malawi, Tanzania, and Zambia [16]. From January 2018 to January 2020, surgical teams at ten district hospitals in Zambia's Southern and Lusaka provinces received quarterly visits from specialist surgeons, anesthetists, and theatre nurses. In addition, a mobile phone-based consultation network was set up for real-time consultations on complex surgical cases and referrals. The primary aim of SURG-Africa is to engage local stakeholders and develop, implement and evaluate a sustainable, locally owned mentoring program [16].

Guided by the principles of the dynamic sustainability framework the SURG-Africa team adopted an approach based on stakeholder involvement and systems science [17–19]. The

dynamic sustainability framework roughly delineates two major challenges that interventions must face: implementation and sustainability. Both are expected to be achieved in constantly changing contexts, necessitating continuous monitoring and adaptation [17]. In designing and supporting implementation of the surgical mentoring model in Zambia, several participatory action research workshops were organized to optimize the fit between the intervention and the local settings through evaluation and adaptation [20]. These workshops involved various stakeholders and used previously collected data from the field. SURG-Africa funding for mentoring trips ended in early 2020, as planned. For it to be continued and become embedded into the Zambian health care system, a suitable regulatory framework is critical. The aim of this paper is, therefore, to utilize lessons learned to explore policy options for embedding surgical mentoring in national health policy through a participatory and dynamic modeling approach, adapted from the field of system dynamics called group model building [21, 22].

## Methods

### Group model building workshops

In February 2020, we held four one-day group model building (GMB) workshops. Two of these focused on central hospitals: Livingstone Central hospital (LCH) and the University Teaching Hospital (UTH) in Lusaka. The two other workshops had their focus on district hospitals (DH): Zimba mission hospital and Namwala district hospital. Participants in the GMB workshops were employed at either health centers, district hospitals, central hospitals, or district/provincial level government departments. This study was reviewed and approved by the University of Zambia Biomedical Research Ethics Committee (approval reference number 528–2019); the Zambian National Health Research Authority granted us authority to conduct the study (approval letter dated 26th December 2019). Prior to the sessions, participants were sent written information about the purpose of the GMB meeting. At the start of each meeting, before the actual group model building was started, the moderator invited participants to share their hopes and fears (or worries) by writing these down on small cards which were displayed on a separate board in the room. After a short group discussion the participants provided their verbal informed consent to participating in the study. This was witnessed by the SURG-Africa country coordinator and recorded in the written notes of each of the four meetings.

Each workshop was started with introductions from the host institution (usually the most senior participant) and the research team. The aim of the day was explained and the group was encouraged to share and be respectful of other participants' views. Workshops were organized using 'scripts', i.e. established procedures for GMB developed over time by the system dynamics community [23–25]. The main script we used was called *Initiating and Elaborating a Causal Loop Diagram*. Workshop participants were introduced to a simple diagram that explained the purpose and ideas behind GMB (Fig 1). The starting diagram consisted of two core factors: "volume of surgery at the DH level" and "surgical mentoring". Construction of the diagram occurred in four phases, focusing on: 1) the requirements of district-level surgical scale-up, 2) its consequences, 3) the requirements of sustainable mentoring, and 4) its consequences. Each group went through all four phases, but the duration of each phase was catered to the groups' expertise and experience. The DH groups consisted mostly of local clinical staff–recipients of surgical mentoring–and health service administrators, and thus the focus was mainly on surgical scale-up. The central hospital (CH) groups consisted mainly of senior staff involved in mentoring, along with administrators from the provincial health offices. This directed the focus to the sustainability of the surgical mentoring intervention.

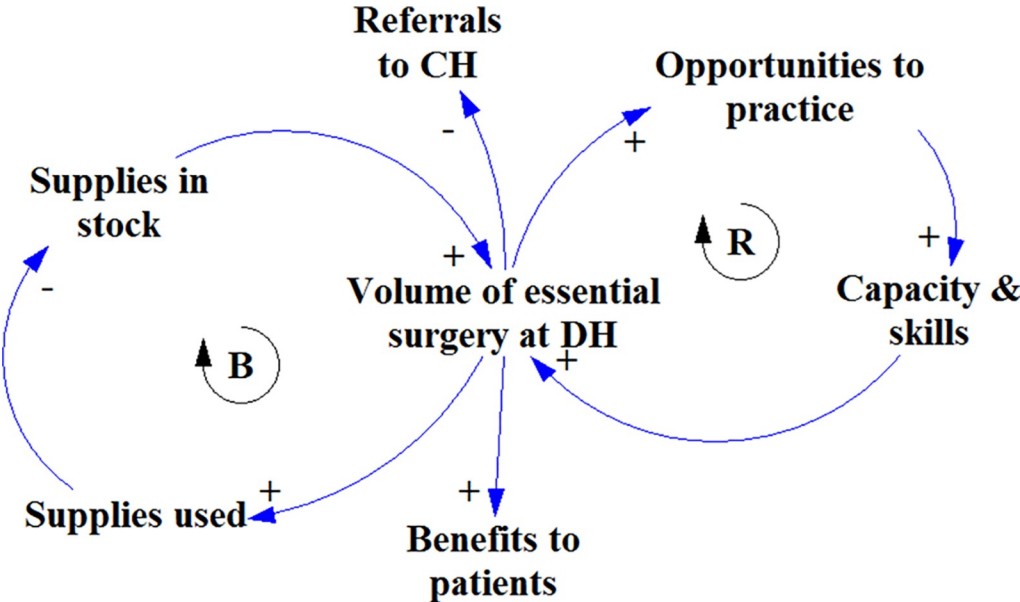

**Fig 1. Simple causal loop diagram that was used as a starting point in the first three group model building workshops.** In addition, it was used to explain the concepts of positive/negative relationships and reinforcing/balancing feedback loops. Plusses indicate relationships were factors change in the same direction, e.g. if volume of surgery goes up, supplies used goes up and if volume of surgery goes down, supplies used goes down. Minuses indicate negative or inverse relationships were factors changes in the opposite direction. So when volume of surgery goes up referrals decrease, and when volume of surgery goes down referrals increase. 'B' indicates a self-correcting or balancing feedback loop. 'R' indicates a self-reinforcing feedback loop.

To increase the quantity and quality of contributed ideas, we also used the *nominal group technique* script during each phase [26]. Participants were given some time to reflect before writing down factors that influenced or were influenced by a central factor. Each participant was given the chance to present and explain one of the factors they had written down. This factor was then, if there was consensus in the group on its inclusion and wording, added to the diagram which was projected on a screen visible to all. This was followed by a group discussion on how the factor might be related to other factors already in the diagram. If there was consensus on a certain causal relationship, this was indicated in the diagram by an arrow. After all participants had contributed at least one factor, a facilitated group discussion was held. During this discussion, factors and their relationships to core or other factors in the diagram could be added, removed, or reworded. Each of the GMB workshops eventually resulted in a completed causal loop diagram (CLD).

## Cleaning and integration of causal loop diagrams

After each workshop, the research team reviewed the CLDs for logical consistency and compared them with the meeting minutes. Where discrepancies were found, the minutes were assumed to best reflect the contributions of participants and the CLD was adjusted accordingly. The final CLDs were communicated to participants after the workshops for internal validation along with a narrative.

Two independent reviewers, HB (expert in health economics and health systems) and MI (medical doctor and expert in health economics), compared the CLDs in a pairwise fashion and identified factors whose meaning matched. Differences in identified matches were discussed, and a third reviewer (LB) was consulted in cases of disagreement. An integrated CLD

was then constructed using clusters of matched factors as connections. Clusters could consist of more than two factors: if factor A is related to factor B and factor B is related to factor C, then all three of them will be in the same cluster. Each cluster was given a label according to the constituent factor that had the most relations with other factors in the cluster. After reviewing the labels, we customized some of them to better reflect cluster contents and/or reduce the length of the label for clarity purposes.

For each link in the final CLD we counted how many times it was mentioned across all four CLDs. The number of times a link between factors $i$ and $j$ was mentioned was denoted as $n_{ij}$. Its value ranged from 1 (i.e. mentioned in one GMB workshop) to 4 (i.e. mentioned in all four GMB workshops). Then, the plausibility $p_{ij}$ of the link between $i$ and $j$ was calculated as:

$$p_{ij} = \begin{matrix} n_{ij}/4, & \text{if link between } i \text{ and } j \text{ positive} \\ -n_{ij}/4, & \text{if link between } i \text{ and } j \text{ negative} \end{matrix} \tag{1}$$

### Structural analysis

As causal loop diagrams are directed cyclic graphs, we used graph theory for analysis [27]. For each factor in both the base and integrated CLD we calculated the following metrics. The *degree* of a factor is the number of connections it has. The *centrality* of a factor is an indicator of its importance in the CLD [28]. We identified all feedback loops (i.e. cyclic paths) in the CLDs. For each we ascertained whether it was *self-reinforcing* or *self-correcting*. Self-reinforcing feedback loops exhibit exponential growth while self-correcting feedback loops seek an equilibrium state. How the system as a whole reacts to change is determined by the interaction of all feedback loops.

Based on the input from GMB participants and the structural analysis, we identified factors that could have policy relevance to sustain surgical mentoring. *Policy levers* or *enablers* are those factors that (most) positively impact surgical mentoring and that can reasonably be expected to be influenced at provincial or central level. *Policy risks* or *barriers* on the other hand are factors that (most) negatively affect surgical mentoring. We used the following metrics as an indication of influence. First, we calculated for each factor the number of simple paths it had leading to surgical mentoring in the integrated CLD, i.e. excluding those paths that included a factor more than once. The plausibility $p$ of path $k$ with links $ij$ in $E$ was calculated by multiplying the plausibility scores:

$$p_k = \prod_{ij \in E} p_{ij} \tag{2}$$

Formula 2 implies that short paths with often mentioned links have a high plausibility $p_k$ while long paths with rarely mentioned links have a low plausibility $p_k$. For example, a path with two links, one with a plausibility of 0.25 and one with a plausibility of 0.75 will have a total plausibility of 0.25·0.75 = 0.165. Secondly, we calculated the total positive, negative, and net plausibility of factors on the core variables. The total positive impact of factor $k$ on surgical mentoring was calculated as

$$I_k^+ = \sum_{p_k > 0, k \in K} p_k \tag{3}$$

Its total negative impact was calculated as

$$I_k^- = \sum_{p_k < 0, k \in K} p_k \tag{4}$$

In both equations, $K$ is the set of all simple paths from factor $k$ to surgical mentoring. The net impact was then calculated as

$$I_k^{net} = I_k^+ + I_k^- \tag{5}$$

The final policy relevant factors are *monitoring* factors, which are those that are important and measurable indicators of the process of sustaining surgical mentoring. The uncertainty $U_k$ of a factor's impact on the core factors was calculated as the percentage of plausibility that pointed in the same direction as its most prevalent direction: We multiplied this by the absolute impact on mentoring:

$$U_k = I_k^{net} \cdot \frac{\max(I_k^-, I_k^+)}{I_k^{net}} \tag{6}$$

## Dynamic analysis of causal loop diagram

A limitation of the structural analysis is that it ignores the (combined) impact of feedback loops. A factor may set a reinforcing loop in motion which over time has large consequences. This may not be evident from its initial (potentially small) influence as seen in the structural analysis. We therefore applied the method to analyse relations between variables using enriched loops (abbreviated as MARVEL) method to explore the CLDs' response to external influences or interventions [29]. For this we modelled the CLD as a set of differential equations, one for each factor. Changes in a particular factor will be propagated through the CLD, 'carried along' the links between factors. In this dynamic analysis we focused on how volume and quality of district-level surgery would change in response to changes in other factors. We limited the dynamic analyses to factors and links that were mentioned in at least two GMB workshops. The effect of a change in factor $i$ (denoted with $\Delta i$) at time $t$ on a factor $j$ where $P_{ij} \neq 0$ (i.e. they are directly linked), was calculated as:

$$v_j(t) = v(t-1) + \Delta i \cdot p_{ij} \cdot s_{ij}^{t-1}(1 - s_{ij}) \tag{7}$$

Please note that this equation represents the effect of only one particular link. In the dynamic analysis, all links were taken into account simultaneously and the value of any factor $j$ at time $t$ depended on the sum of changes in any of the factors that affect it (i.e. where $p_{ij} \neq 0$) at time $t-1$. We defined several factors as being rate-limited which meant that their value depended on their *lowest* input instead of on the sum: volume of surgery, procurement of supplies, and diagnostic skills at a health center (HC). The speed constants $s_{ij}$ of each link were defined independently by two researchers (HB and MI) and expressed as a score on a scale of 0 to 1; in the final model we took the mean of the two assigned scores. To explore the impact of each factor we introduced a constant change of 1 for time period $t = 1$ until $t = 100$ and measured how all other factors in the CLD changed over time until $t = 1000$. We measured the mean change of over the time-period as a response to these changes and used the standard deviation of this change as an indicator of uncertainty/instability.

**Table 1. Composition (cadre and number) of participants to the group model building workshops.**

| Cadre | Zimba District | Namwala | LCH | UTH | Total |
|---|---|---|---|---|---|
| Surgeons | - | - | 2 | 1 | 3 |
| Anesthesiologists | - | - | 1 | 1 | 2 |
| Hospital administrators (MOI, Matron, accountant, or HR) | 4 | 3 | - | 3 | 10 |
| District health managers (DHD, accountants, or planners) | 6 | 3 | - | - | 9 |
| Provincial health managers | - | - | 3 | - | 3 |
| Medical officers | 1 | - | | - | 1 |
| Medical Licentiates | - | - | 1 | | 1 |
| (Theatre) Nurses | 4 | 2 | 3 | 2 | 11 |
| Anesthetists | | 1 | 1 | 1 | 3 |
| Clinical officers (mostly health center managers) | 3 | 5 | - | - | 8 |
| Unknown (Incomplete data) | - | - | 1 | - | 1 |
| SURG-Africa researchers (excluding facilitation team) | 1 | 1 | 3 | 4 | 9 |
| **Total** | 19 | 15 | 15 | 12 | **61** |

LCH = Livingstone Central Hospital, UTH = University Teaching Hospital, MOI = medical officer in-charge, HR = human resources, DHD = district health directorate.

## Results

### Group composition

The total number of participants over the four workshops was 61 (Table 1). With 12 participants, the group in Lusaka was the smallest. Both workshops at CHs included specialists and had no representatives from health centers. The group in Livingstone was the only group in which provincial representatives were present. At the DHs, the groups consisted mainly of DH staff, district health officials, and representatives from health centers. Many participants with administrative duties also had a clinical background, e.g. among the hospital managers in-charges are usually medical doctors and sometimes medical licentiates. Most district health directors also had a medical background.

### Causal loop diagrams

In the diagram built at the Livingstone CH workshop (Fig 2), the most prominently mentioned factor was the establishment of a 'mentoring faculty' (i.e. an assigned team or pool of mentors) with zonal focal points–a person/desk at the provincial level that would be responsible for planning and coordinating surgical mentoring. Possibilities to integrate surgical mentoring in existing health sector programs and budgets for technical support and supervision were also mentioned. The participants did not link surgical mentoring to surgical volume or quality at the district level, but it was mentioned that it could incentivize staff to take up posts at the DH; which would increase staff motivation and the quality of surgery.

The Namwala DH diagram (Fig 3) focused mainly on factors surrounding surgery at the DH itself. Integrating surgical mentoring with existing mentorship programs was the only enabler mentioned for surgical mentoring. The group did mention a direct link between surgical mentoring and their capacity and skills as a district-level surgical team. Existing mobile phone-based surgical consultation networks were also reported as being beneficial to capacity and skills; among the prerequisites for its effectiveness were learning from other intra-district mobile consultation experiences and the contribution of a good moderator [30].

At Zimba Mission hospital (Fig 4), a link was made between patient demand for surgery and the DH's priority for surgery. In addition, to sustain surgical mentoring, MOH and

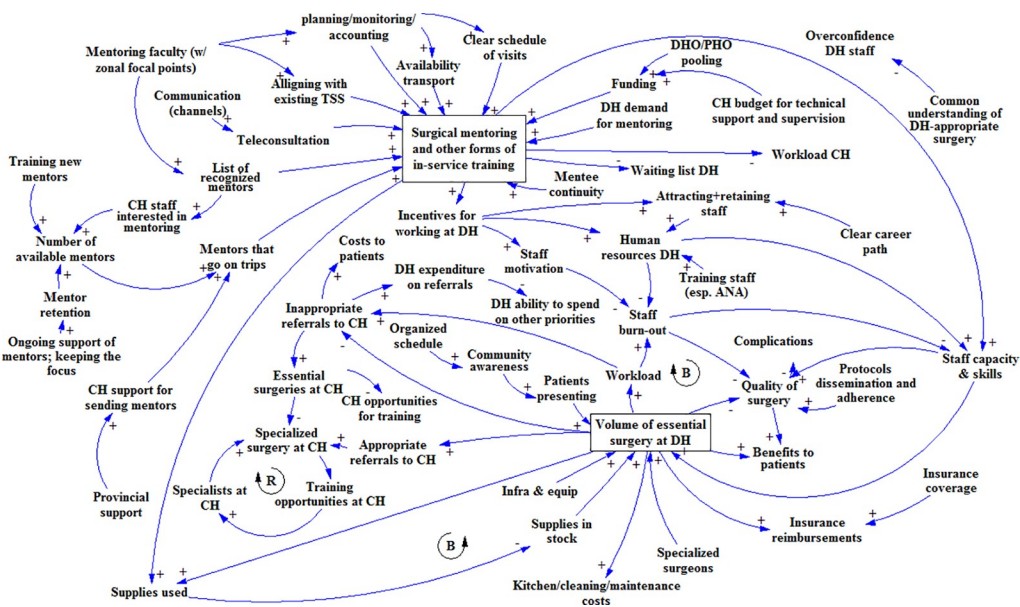

**Fig 2. Causal loop diagram as it was built in the first GMB workshop in Livingstone.**

provincial support were deemed necessary; and cost-effectiveness evidence was seen as a way to gain such support. Mentee (staff) turnover was mentioned as a risk to surgical mentoring, caused mainly by transfers and non-availability of staff who had left for further training: e.g. freshly graduated medical doctors need to do two years of rural posting to be able to specialize, and nurses rotate frequently. Acknowledgments and certification of mentees was mentioned as a factor that could reduce mentee turnover.

Because the UTH Lusaka workshop group was relatively small and consisted mostly of senior staff, it was decided to start the GMB session with a CLD of surgical scale-up, based on the output from the three workshops held earlier. This allowed the participants to double-check the validity of the earlier diagrams, while enabling the moderators to focus the

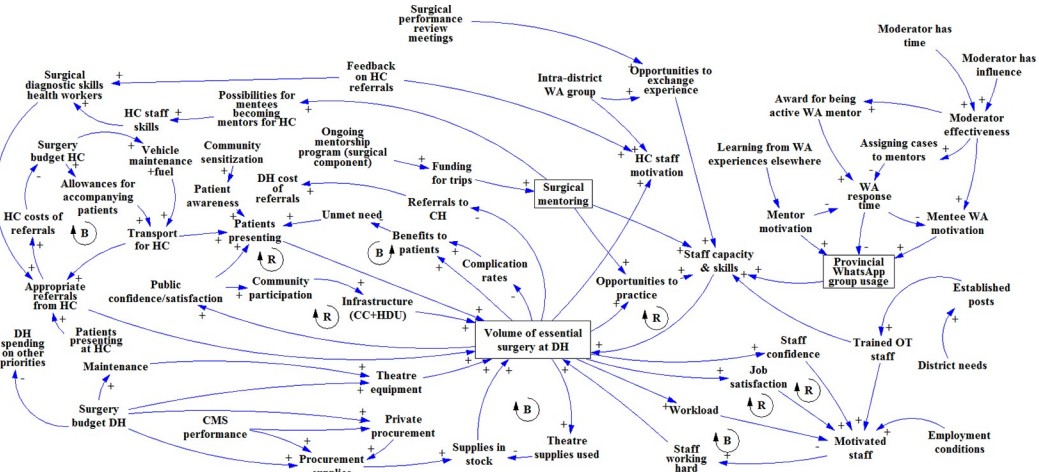

**Fig 3. Causal loop diagram as it was built in the GMB workshop at Namwala district hospital.**

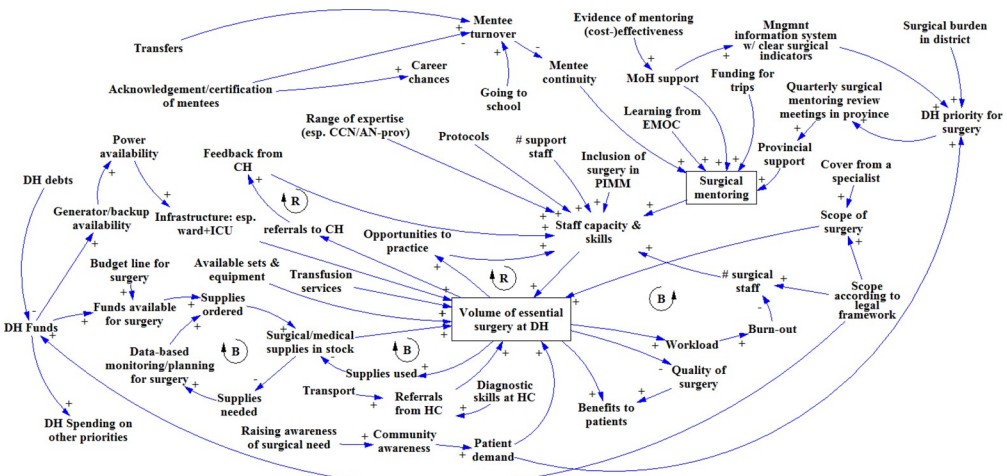

**Fig 4. Causal loop diagram as it was built in the GMB workshop at Zimba Mission hospital.**

discussion on the mentoring aspect (rather than on DH-level surgical scale-up), with which the participants had much experience. The CLD at the start of the workshop did not include the mentoring part, so as not to bias the group discussion. The participants added factors concerning self-referrals from central hospitals, DH funds, and the use of untrained staff (Fig 5). They argued that surgical mentoring and support required the involvement of provincial (health) authorities and professional associations, and again cost-effectiveness data on surgical mentoring was mentioned as an enabler. In addition, they suggested that existing mentorship guidelines be enforced and that mentoring be included in DH and CH annual action plans. These elements would increase the opportunities specialists have to undertake effective surgical mentoring.

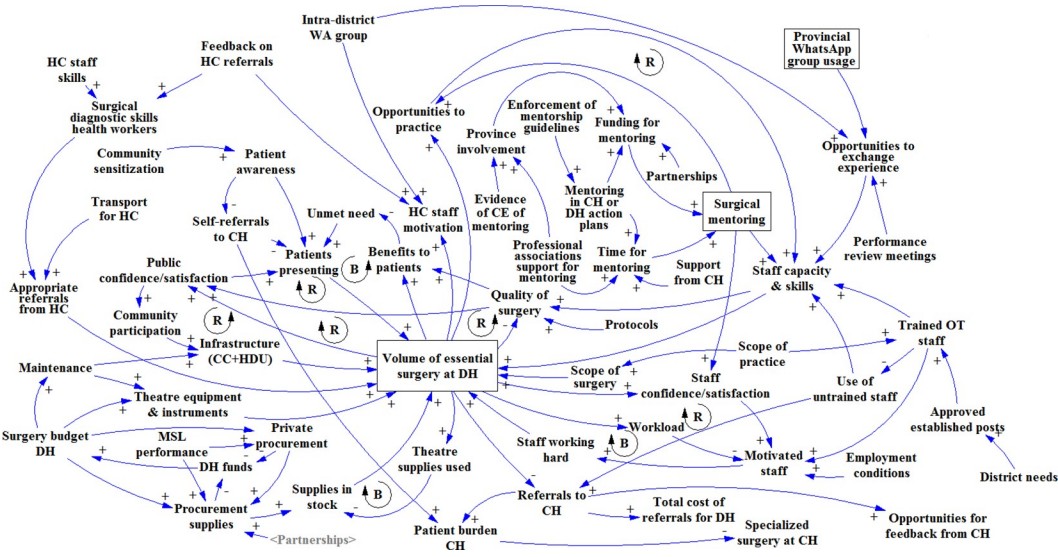

**Fig 5. Causal loop diagram as it was built in the GMB workshop in Lusaka.**

**Table 2. Number of matching factors identified in the pairwise causal loop diagram comparisons, with average overlap in between brackets (relative to the total number of factors in the CLDs).**

| | Livingstone CH | Namwala DH | UTH Lusaka | Zimba MH |
|---|---|---|---|---|
| Livingstone CH | - | 23 (38%) | 28 (47%) | 29 (51%) |
| Namwala DH | - | - | 43 (72%) | 28 (49%) |
| UTH Lusaka | - | - | - | 36 (63%) |
| Zimba MH | - | - | - | - |

Only numbers above the diagonal shown for clarity. CH = central hospital, DH = district hospital, UTH = university teaching hospital, MH = mission hospital.

### Integrated causal loop diagram

In total 187 matches were found across the six pairwise comparisons of four GMB workshop results. Agreement between the two reviewers on what constituted a match was the lowest in comparing Livingstone CH with the two DHs (54% and 58%), and highest in comparing Namwala DH to UTH Lusaka (95%). Table 2 shows the number of agreed-on matches per pairwise CLD comparison along with the final percentage overlap. The number of matches was highest between the CLDs of Namwala and UTH Lusaka (43, or 72%), and lowest between the CLDs of Livingstone CH and Namwala DH (23, or 38%).

The total number of clusters identified through the CLD was 52. The largest cluster (*human resource management* or *HRM* in short) consisted of 11 factors, including factors such as *staff capacity and skills*, *trained operating theatre staff*, and *human resources DH*. The second largest cluster, *aligning with existing technical support and supervision*, consisted of factors relating to currently running supervision efforts, inclusion of mentoring in local/provincial action plans, and the more general factor *provincial support*. The smallest clusters (of which there were 24) were simple pairs of factors, most of which were identically worded.

The resulting integrated CLD consists of 119 factors (Fig 6). When we limit the CLD to factors mentioned at least 2, 3, or 4 times the CLD size reduces to 45, 22, and 12, respectively. The graph-theoretic metrics for the 25 most connected factors are presented in Table 3. The mean degree across the CLD is 3.27 (95% confidence interval: 1.00 to 8.45). For *volume of surgery DH* the total degree is the highest at 22, followed by *surgical mentoring* with 21 and *HRM* with 17. Across the CLD, 30 factors have a degree of 1 (i.e. they are at the boundary of the graph). Of these, only five have a link coming in, with 25 having a link going out. The centrality corresponds roughly with the degree (correlation is 0.84). Again, *volume of surgery DH*, *surgical mentoring*, and *HRM* have the highest centrality. However, there are some factors that have a high centrality although their degree is lower, e.g. *DH priority for surgery* or *quarterly surgical mentoring review meetings in province*. This mismatch is due to these factors being closely related to high-centrality.

### Structural analysis

Tables 4 and 5 shows the results from the structural and dynamic analyses of impact on surgical mentoring, respectively. In the structural analysis, the factors that seem most likely to increase the sustainability of surgical mentoring are both institutional and practical. Influential institutional support factors were identified at different levels: national (i.e. MOH), provincial medical office, and professional (medical and NPC) associations. Influential practical enablers were funding, mentee continuity, and having a list of recognized mentors.

There are two broad categories of policy risks or barriers for surgical mentoring. Firstly, there are factors relating to mentees not being available at the DH, either because of transfers

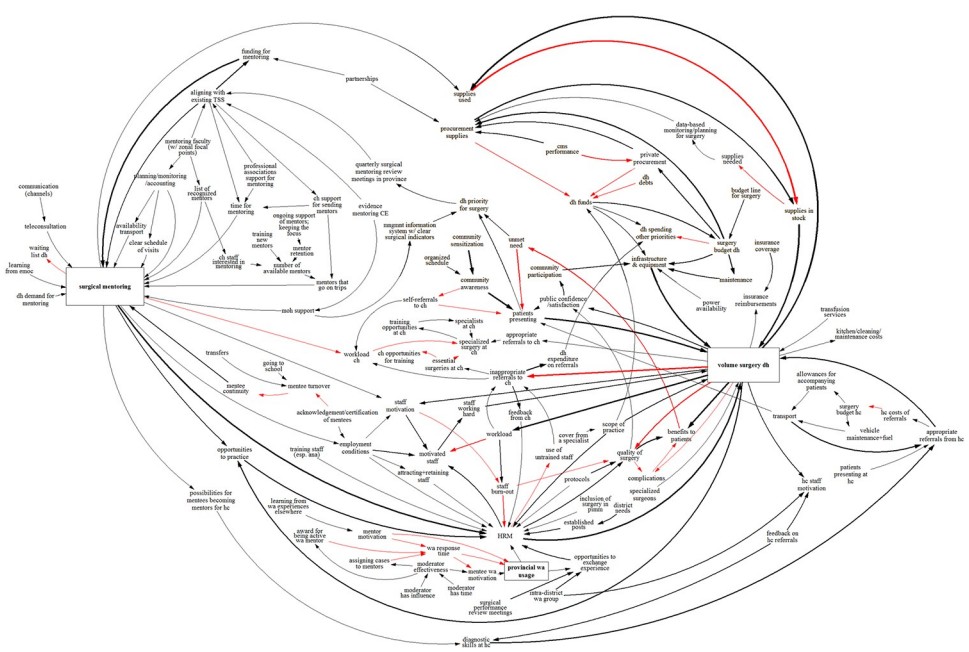

**Fig 6. Integrated CLD with all matched factors.** Black arrows denote positive links, while red arrows denote negative links. The thickness of arrows indicates how often links were mentioned across workshops. The boxes indicate the core factors. As there are many feedback loops the notation for them has been omitted for clarity.

**Table 3. Graph-theoretic metrics for the integrated causal loop diagram.**

| | Centrality score | Degree in (*causes*) | Degree out (*effects*) |
|---|---|---|---|
| volume surgery DH | 3346 | 9 | 13 |
| surgical mentoring | 2393 | 13 | 8 |
| HRM | 2326 | 13 | 4 |
| aligning with existing TSS | 1395 | 4 | 4 |
| dh priority for surgery | 1393 | 3 | 1 |
| quarterly surgical mentoring review meetings in province | 1351 | 1 | 1 |
| patients presenting | 1182 | 5 | 2 |
| supplies in stock | 944 | 2 | 2 |
| public confidence/satisfaction | 900 | 2 | 2 |
| appropriate referrals from HC | 802 | 3 | 2 |
| benefits to patients | 752 | 3 | 1 |
| inappropriate referrals to CH | 712 | 3 | 5 |
| unmet need | 710 | 1 | 2 |
| supplies used | 709 | 2 | 1 |
| diagnostic skills at HC | 668 | 2 | 1 |
| possibilities for mentees becoming mentors for HC | 651 | 1 | 1 |
| procurement supplies | 604 | 5 | 2 |
| quality of surgery | 601 | 4 | 3 |
| supplies needed | 549 | 1 | 1 |
| provincial WhatsApp usage | 549 | 3 | 2 |

For clarity only those factors that are in the top 20 highest centralities are shown here. DH = district hospital, HRM = human resource management, TSS = technical support and supervision, HC = health center.

**Table 4. Results from the qualitative structural analysis to investigate impact on surgical mentoring.**

| | Factor name | | Links to surgical mentoring | | | | Plausibility of links | | | |
|---|---|---|---|---|---|---|---|---|---|---|
| | | n | Shortest | Longest | Neg | Pos | Neg | Pos | Net | Uncertainty |
| Top 10 enablers | *Aligning with existing TSS* | 5 | 2 | 4 | 0 | 5 | - | 1.34 | 1.34 | - |
| | *funding for mentoring* | 1 | 2 | 2 | - | 1 | - | 1.00 | 1.00 | - |
| | *mentee continuity* | 1 | 2 | 2 | - | 1 | - | 0.50 | 0.5 | - |
| | *mentoring faculty (w/ zonal focal points)* | 10 | 3 | 6 | - | 10 | - | 0.49 | 0.49 | - |
| | *evidence mentoring CE* | 11 | 3 | 9 | - | 11 | - | 0.40 | 0.40 | - |
| | *professional associations support for mentoring* | 6 | 3 | 5 | - | 6 | - | 0.40 | 0.40 | - |
| | *planning/monitoring/accounting* | 3 | 2 | 3 | - | 3 | - | 0.38 | 0.375 | - |
| | *quarterly surgical mentoring review meetings in province* | 5 | 3 | 5 | - | 5 | - | 0.37 | 0.37 | - |
| | *MOH support* | 6 | 2 | 8 | - | 6 | - | 0.26 | 0.26 | - |
| | *list of recognized mentors* | 2 | 2 | 5 | - | 2 | - | 0.25 | 0.25 | - |
| Top 10 barriers | *mentee turnover* | 1 | 3 | 3 | 1 | - | 0.13 | - | -0.13 | - |
| | *Transfers* | 1 | 4 | 4 | 1 | - | 0.03 | - | -0.03 | - |
| | *going to school* | 1 | 4 | 4 | 1 | - | 0.03 | - | -0.03 | - |
| | *benefits to patients\** | 10 | 6 | 9 | 10 | - | 0.02 | - | -0.02 | - |
| | *quality of surgery\** | 55 | 7 | 15 | 50 | 5 | 0.02 | - | -0.01 | 0.15 |
| | *HRM\** | 255 | 8 | 17 | 150 | 105 | 0.03 | 0.02 | -0.01 | 0.40 |
| | *Protocols\** | 310 | 8 | 18 | 200 | 110 | 0.02 | 0.01 | -0.01 | 0.30 |
| | *scope of practice\** | 1605 | 8 | 22 | 870 | 735 | 0.03 | 0.03 | -0.01 | 0.43 |
| | *opportunities to practice\** | 255 | 9 | 18 | 150 | 105 | 0.02 | 0.02 | -0.01 | 0.40 |
| | *surgery budget DH\** | 1575 | 9 | 21 | 795 | 780 | 0.04 | 0.03 | -0.01 | 0.46 |
| Top 10 uncertain | *CMS performance* | 2700 | 10 | 23 | 1380 | 1320 | 0.02 | 0.02 | 0.00 | 0.48 |
| | *Transport* | 320 | 6 | 18 | 170 | 150 | 0.02 | 0.02 | 0.00 | 0.48 |
| | *procurement supplies* | 900 | 9 | 21 | 435 | 465 | 0.02 | 0.02 | 0.00 | 0.47 |
| | *data-based monitoring/planning for surgery* | 900 | 10 | 22 | 435 | 465 | 0.01 | 0.01 | 0.00 | 0.47 |
| | *surgery budget DH* | 1575 | 9 | 21 | 795 | 780 | 0.04 | 0.03 | 0.01 | 0.46 |
| | *budget line for surgery* | 1575 | 10 | 22 | 795 | 780 | 0.01 | 0.01 | 0.00 | 0.46 |
| | *volume surgery DH* | 225 | 7 | 16 | 120 | 105 | 0.02 | 0.02 | 0.00 | 0.45 |
| | *supplies in stock* | 900 | 8 | 24 | 480 | 420 | 0.02 | 0.02 | 0.00 | 0.45 |
| | *infrastructure & equipment* | 225 | 8 | 17 | 120 | 105 | 0.02 | 0.02 | 0.00 | 0.45 |
| | *DH funds* | 1125 | 9 | 21 | 600 | 525 | 0.02 | 0.02 | 0.00 | 0.45 |

Neg = negative, Pos = positive, TSS = technical support and supervision, CE = cost-effectiveness, MOH = ministry of health, HRM = human resource management, DH = district hospital, CMS = central medical stores.

*these factors are no longer barriers when decreases in 'DH priority for surgery' are no longer coupled to the efficacy of surgical mentoring.

or because they are absent pursuing further training. The second category of barriers are, counterintuitively, factors that are beneficial such as *benefits to patients* and *quality of surgery*. Of course, such inherently beneficial factors should not be labeled as barriers. By carefully observing the CLD, it can be seen that the reason for these results is the factor *DH priority for surgery*. If its effects are traced through the CLD, the following dynamics can be observed: when fewer patients present because the surgical needs in the community are met as a result of good quality surgery (an effect of mentoring), the DH's priority for surgery would decrease. This may decrease DH involvement in quarterly surgical review meetings and subsequently the effectiveness of surgical mentoring.

When the link between DH priority for surgery and its involvement in surgical review meetings is removed, the counterintuitive negative effects of these otherwise beneficial factors

**Table 5. Results of quantitative analysis of expected changes in volume of surgery at the district level using the MARVEL method.**

|  | Factor name | Mean change in surgical volume | Standard deviation of change |
|---|---|---|---|
| Top 10 enablers | scope of practice | 33 | 17 |
|  | benefits to patients | 27 | 15 |
|  | staff motivation | 17 | 14 |
|  | protocols | 14 | 10 |
|  | workload | 13 | 10 |
|  | employment conditions | 12 | 10 |
|  | opportunities to practice | 11 | 9 |
|  | HRM | 11 | 9 |
|  | feedback from CH | 10 | 9 |
|  | established posts | 10 | 9 |
| Barriers | staff burn-out | -37 | 17 |
| Top 10 uncertain | inappropriate referrals to ch | 34 | 18 |
|  | staff burn-out | -37 | 17 |
|  | scope of practice | 33 | 17 |
|  | benefits to patients | 27 | 15 |
|  | staff motivation | 17 | 14 |
|  | employment conditions | 12 | 10 |
|  | workload | 13 | 10 |
|  | protocols | 14 | 10 |
|  | established posts | 10 | 9 |
|  | opportunities to exchange experience | 10 | 9 |

TSS = technical support and supervision, CE = cost-effectiveness, MOH = ministry of health, HRM = human resource management, DH = district hospital,
CMS = central medical stores.

*these factors are no longer barriers when decreases in 'DH priority for surgery' are no longer coupled to the efficacy of surgical mentoring.

disappear. In other words, the DH's priority for surgery and its demand for mentoring is very important to stimulate and monitor closely. This is even more so the case when one considers the fact that DH administrators can to some extent influence mentee motivation and continuity. The most uncertain factors in the qualitative analysis are Central Medical Stores (CMS) performance, transport, and procurement of supplies. Note that in the qualitative analysis there is no uncertainty regarding the impact of the most influential policy risks and policy levers. Overall, the uncertainty around barriers is much higher than the uncertainty around enablers.

## Dynamic analysis of impact on district-level surgery

The mean difference in assigned speed (i.e. the $s_{ij}$'s in Formula 7) between the two reviewers was 0.58 (95% CI: 0.40 to 0.76). In 74% of links, the difference in speed was one or less (Fig 7). Table 5 shows the most influential and uncertain enablers and barriers of district-level volume of surgery. Figs 8–11 shows how a selected set of factors (including the core variables) change in response to a short stimulation of four of these (full list available in the S1 Data). Observations of note are that DH volume of surgery seems to change after a while when only (funding for) surgical mentoring is introduced. This is because mentoring is assumed to involve an increase in the use of supplies and the integrated CLD does not take into account a shared (large) pool of supplies being used simultaneously by mentoring and routine surgery. Also, quality of surgery is expected to quickly improve after starting mentoring but it then decreases

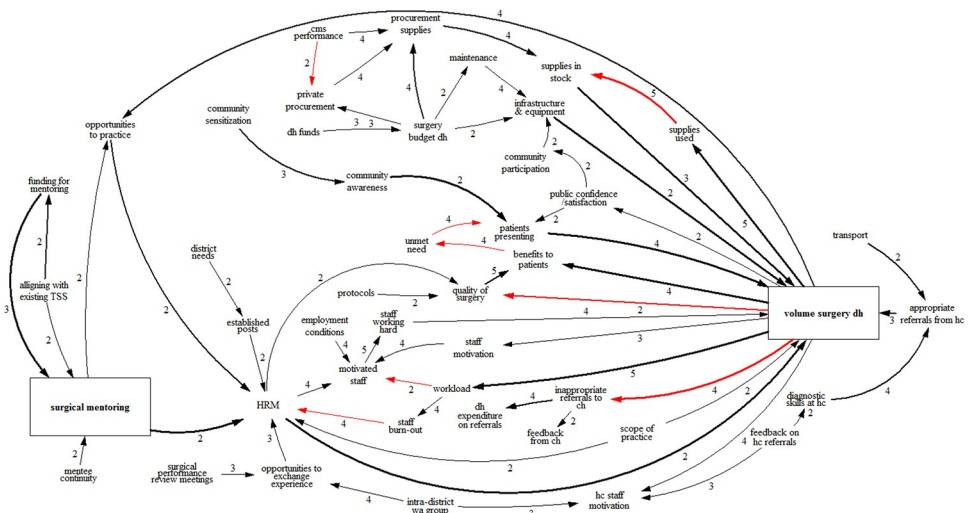

**Fig 7. Integrated CLD with only factors that were mentioned in more than one workshop.** Black arrows denote positive links, while red arrows denote negative links. The thickness of arrows indicates how often links were mentioned across workshops. The boxes indicate the core factors. The numbers beside arrows indicate the speed of change, with 1 being the slowest and 5 being the fastest. As there are many feedback loops the notation for them has been omitted for clarity.

slowly over time due to increased workload. When the scope of surgical practice is expanded, the volume increase is more immediate. There is also an increase in quality of surgery, which is lower than in the case of mentoring, but more stable over time. In addition, the DH's surgical expenditures would be increased over time while scope is expanded. Implementing surgical protocols has an effect similar to mentoring, but the volume increase is achieved earlier. Finally, increased staff burn-out has serious negative effects on both surgical volume and quality.

## Policy recommendations

Several recommendations for sustaining surgical mentoring can be derived from the structural analysis of the integrated CLD and the underlying four stakeholder consultations. Although the previous sections discussed individual factors, any policy decision to continue surgical mentoring will depend on the simultaneous enactment of several measures. The recommendations listed below are illustrated with colors in the integrated CLD as presented in Fig 12.

Ideally the actor responsible for surgical mentoring will have the required mandate and be held accountable by national and/or provincial health authorities. The most likely candidate for such an actor would be a provincial mentoring faculty (PMF), as suggested in the Livingstone CH diagram, which would include senior program managers and relevantly trained (i.e. matching DH needs) surgical specialists. The PMF would be responsible for organizing mentoring trips, mobilizing financial resources from central sources (for example those set aside for NSOAP implementation) and partnerships. For the PMF to be effective it is also critical that it is established in close coordination with professional associations. Such associations can delegate members to be part of the PMF or members can contribute in an advisory role. In addition, professional associations can help in making sure existing clinical mentorship guidelines are used in the context of surgery and assist in developing any required new tools [12]. Being integrated in the provincial health administration, the PMF should seek to integrate surgical mentoring with existing technical support programs (e.g. those for disease control).

## Stimulating `funding for mentoring` until i=100

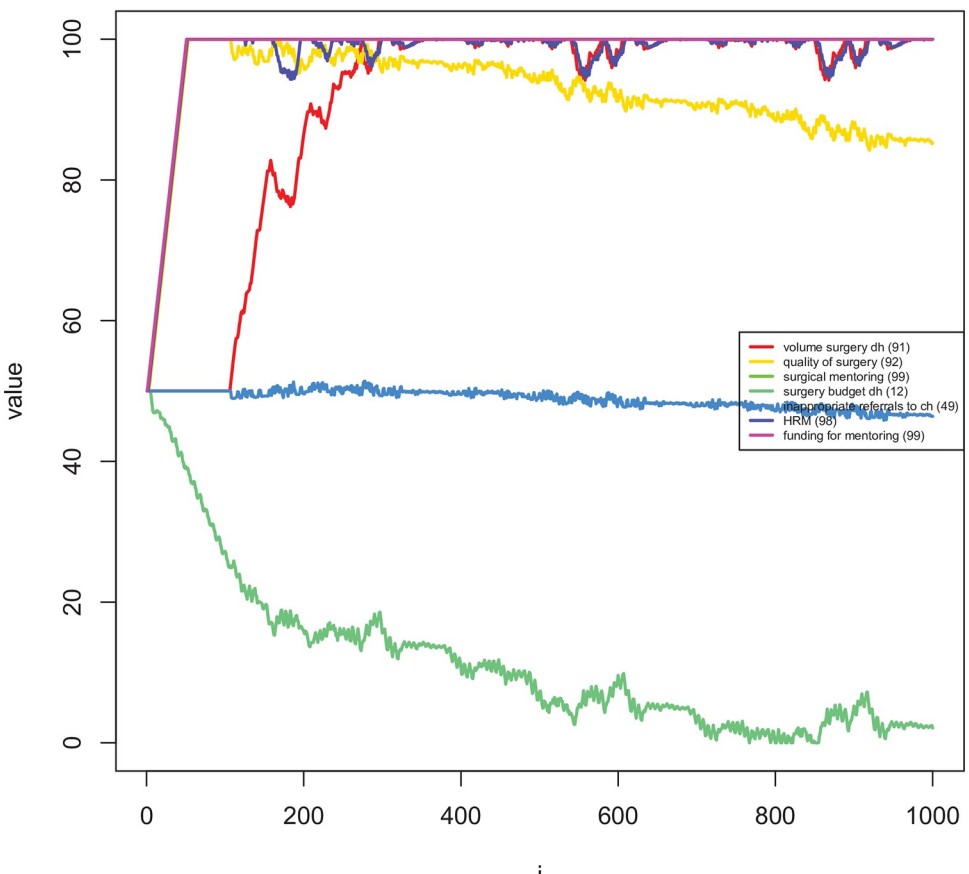

**Fig 8. Long term dynamic explorations of the effects on changes in 'funding for mentoring' for the first 100 time periods.**

Long-term objectives that can help to solidify surgical mentoring are the training of new mentors and certification of mentees, both ideally after guidelines have been adopted.

Apart from the central role of a mentoring faculty, other stakeholders can contribute as well. DH management teams and the PMF have a joint responsibility to ensure that district-level surgical teams are ready to receive mentors. Physical readiness requires that mentoring trips are scheduled in close coordination with DHs who need to ensure that all the necessary supplies are available; for the latter a checklist could be developed to help DHs prepare themselves. In addition, the role of the CMS is important, as any lapse in its performance could result in DHs having to turn to (more expensive) procurement of supplies from the private market.

There should be an early warning system for eroding DH (and CH) support [12]. One way PMFs can do this is by liaising closely with DHs and collecting trip reports from mentors. Mentors should signal whenever a DH turns out not ready to receive them. The PMF should keep a close eye on these reports and (the reasons behind) cancellations of mentoring trips by either mentors or DHs. Ideally there would be quarterly surgical mentoring review meetings at the provincial level, led by the PMF, that include mandatory representatives from DHs, CHs, and professional associations. Such meetings could adapt surgical mentoring as per the need

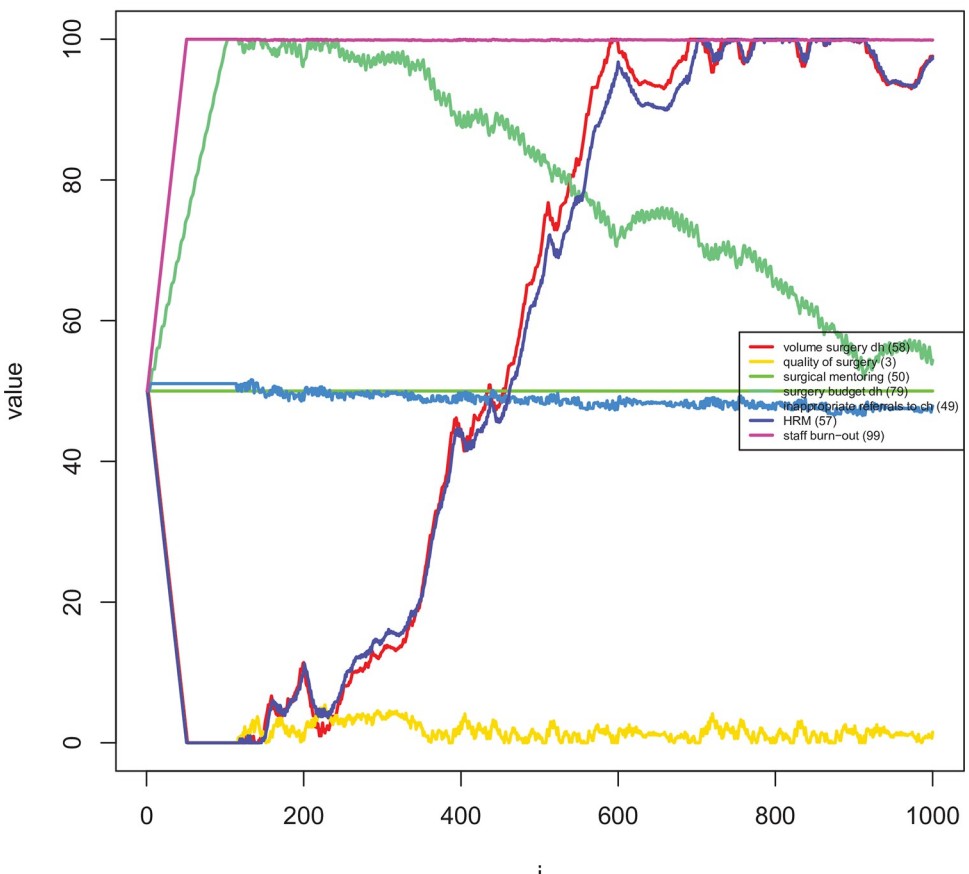

**Fig 9. Long term dynamic explorations of the effects on changes in 'staff burn-out' for the first 100 time periods.**

of relevant stakeholders. Data to substantiate ongoing cost-effectiveness evaluations (costs, volume and quality of surgery) should be collected, as a continuation of what is currently being collected in the SURG-Africa project.

Hospital management teams at DHs would need to foster an atmosphere that is conducive to continuing surgical improvement. They should closely monitor staff motivation and encourage participation of staff in mentoring and internal surgical quality reviews. It should be clear to staff what is expected of them and what they can gain by becoming a mentee. DH management teams would be helped in this if guidelines were available and if certification of mentees is a possibility. In addition, it is important that district hospital management teams can signal if for instance a supply shortage prohibits its ability to receive mentoring teams. A data-based monitoring tool for surgical supplies could be instrumental. For DHs it is also important to have a dedicated budget for surgery, which can cover supplies and any additional resources needed for effective surgical mentoring. District hospitals should be empowered to engage in sensitization efforts so local communities are aware of the timing and types of surgery available at their district hospital. Inclusion of mentoring in annual performance reviews could be a way to embed these activities.

Central hospital management teams may report conflicting priorities, suspecting that the sending out of mentors could negatively affect surgery at their own institutions. It is important

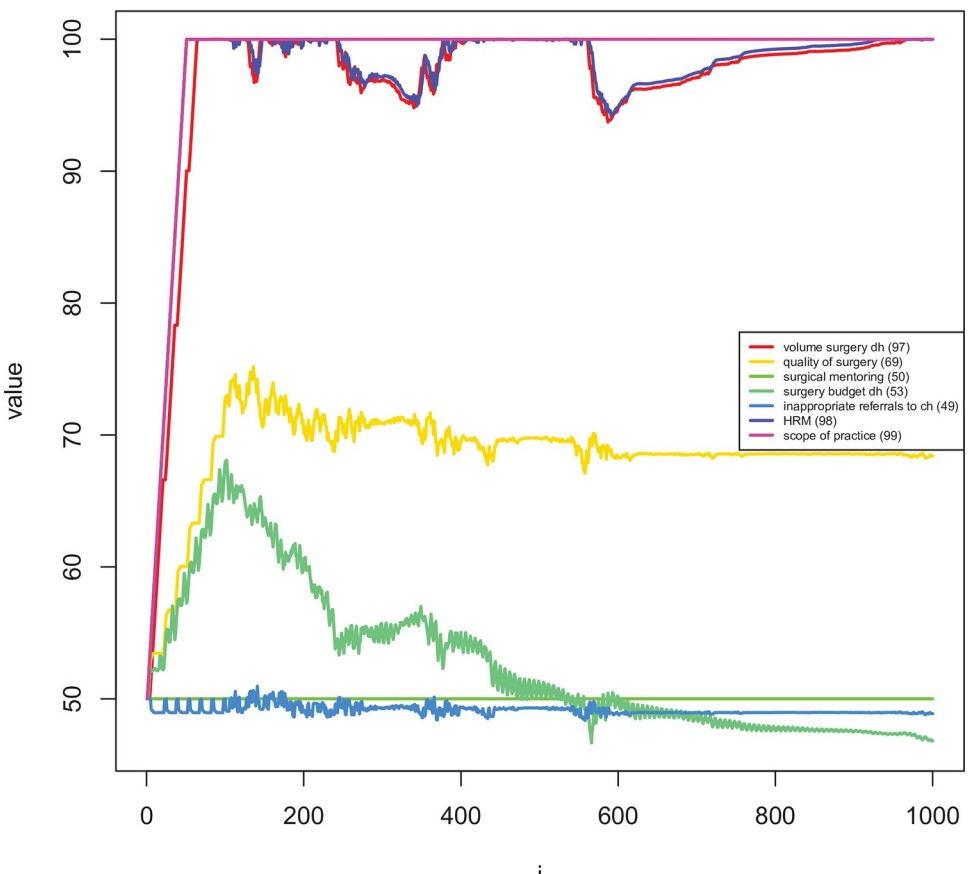

**Fig 10. Long term dynamic explorations of the effects on changes in 'scope of practice' for the first 100 time periods.**

for CHs to keep a long-term perspective and acknowledge that in the long-term mentoring may reduce undue patient referrals as DH teams are empowered to do basic surgery locally. This would help CHs to focus more on specialized surgery.

## Discussion

In this study we have elicited the views of local stakeholders on the sustainability of surgical mentoring in Zambia, using an innovative, systematic, and replicable process. We included perspectives of stakeholders with a variety of backgrounds and with different professional responsibilities using a participatory approach rooted in systems science. Based on stakeholder input and a structural analysis of causal loop diagrams, policy recommendations were developed that hinge on the establishment of a provincial mentoring faculty to support district-level surgery for rural and remote populations.

There is increasing evidence for the (cost-)effectiveness, safety, and feasibility of scaling-up essential surgery at the district level [7, 31]. Surgical mentoring can play a key role in making this happen, but its sustainable implementation is a complex affair. Our results reflect the neglected state of surgery in Zambia, beyond central hospital-level surgery and caesarean

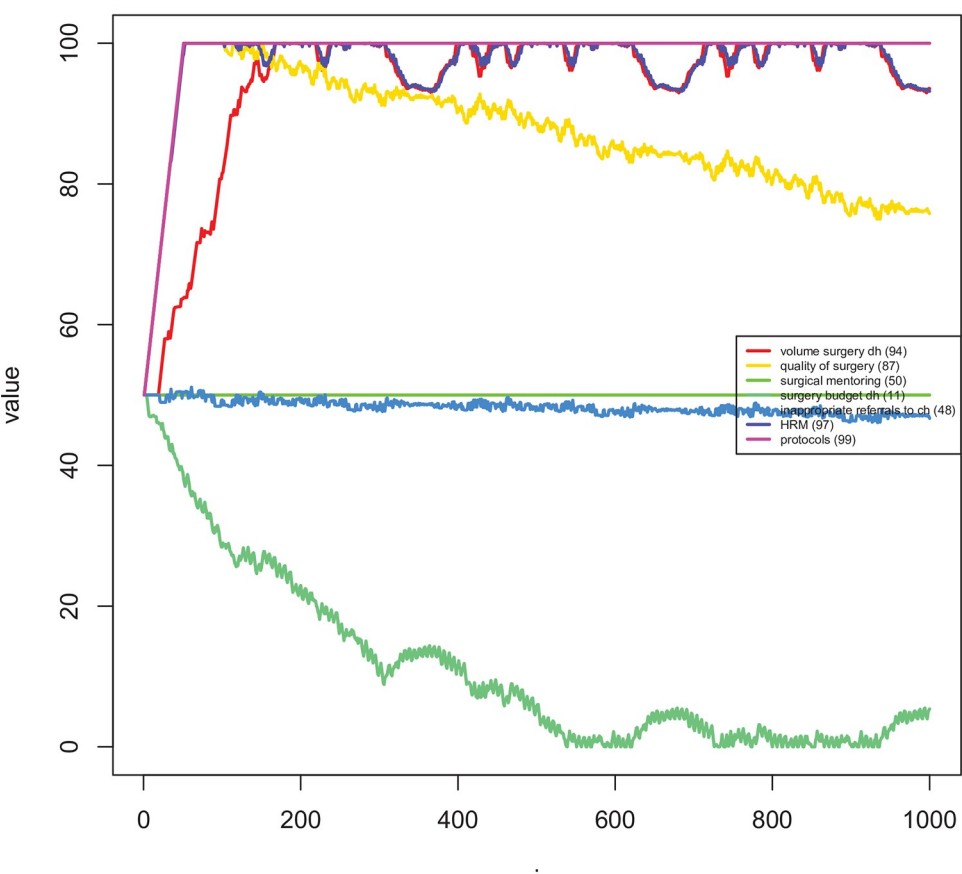

**Fig 11. Long term dynamic explorations of the effects on changes in 'protocols' for the first 100 time periods.**

sections at general hospitals. Intervening in such a complex adaptive system is difficult because some policy levers may not be apparent beforehand, while seemingly beneficial interventions can have unintended consequences [18, 32]. For implementation plans to be effective and sustainable, they need to allow for a constant monitoring of influential factors in the system and allow for continuous adaptation in response to change [17, 33]. An implementation policy without these aspects would decrease the likelihood that mentoring is sustained and in the long term it would make NSOAP aims around district level surgery harder to attain.

Throughout this work we have aimed to develop recommendations consistent with the tenets set out in dynamic sustainability framework literature [17]. Earlier work by the SURG-Africa team has demonstrated stakeholder involvement and continuous learning during the development and implementation of the surgical mentoring intervention [15, 16, 20]. This study focused mainly on the tenet "*Programs should be more likely to be maintained when there is strong 'fit' between the program and the implementation setting*" [17]. This fit was pursued by building on existing relations with stakeholders and employing the GMB method from system dynamics. To the best of our knowledge, this study is the first to employ GMB for this purpose [17, 34]. We would argue that it is a valuable method in the toolbox of implementers trying to apply the (rather theoretical) concepts in the dynamic sustainability framework. Of course, stakeholder interviews and focus groups already have the benefit of involving stakeholders.

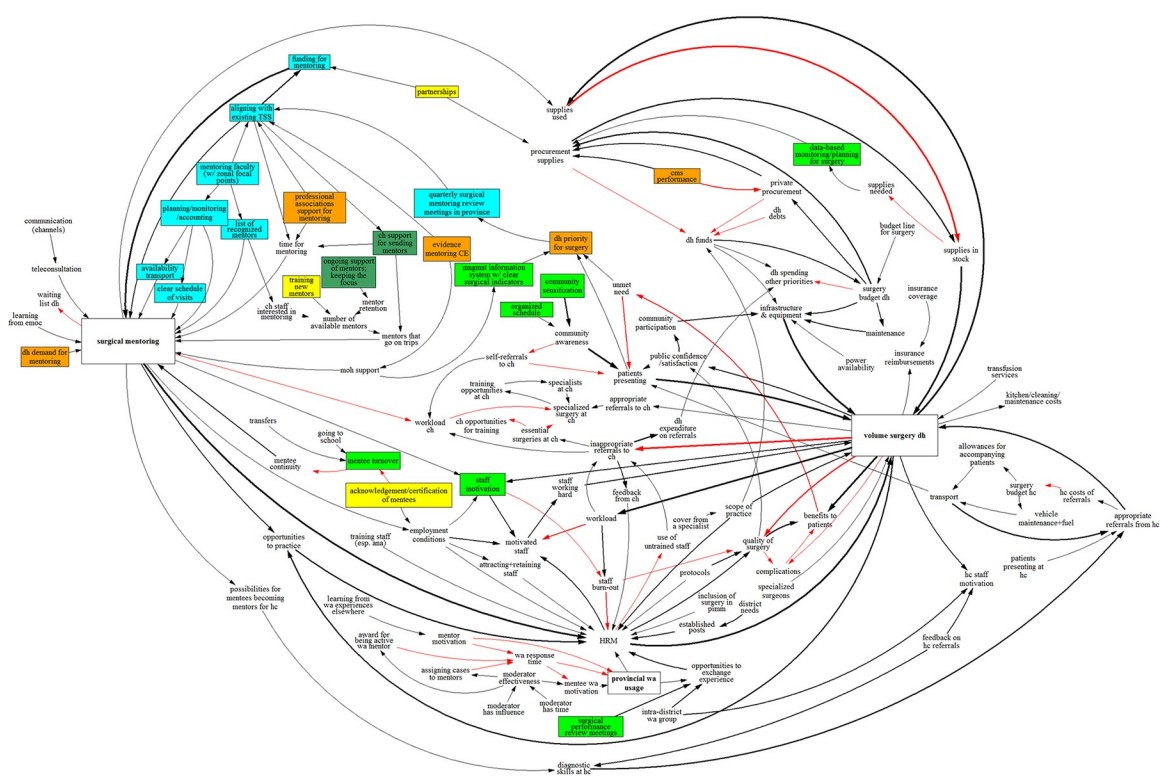

**Fig 12. Integrated causal loop diagram showing the recommendations for sustaining surgical mentoring.** Black arrows denote positive links, while red arrows denote negative links. The thickness of arrows indicates how often links were mentioned across workshops. As there are many feedback loops the notation for them has been omitted for clarity.

However, interviews miss the group-level learning and commitment building effects of GMB [35], and both methods are less well suited to investigating complex systems due to bounded rationality [36].

The primary limitation of our approach is that the results are not, as yet, supported by (clinical) evidence. This puts the results at the same level of confidence as expert opinion, albeit from a perhaps more relevant group, i.e. those who need to implement changes on the ground; and after internal discussion. The results also probably reflect the participants' fields of expertise and work environment. This is shown in the percentage of overlap between CLDs presented Table 1. The overlap between LCH and the DHs was low, but it was high for UTH where the discussion was started based on an expanded diagram that included the results from previous workshops. Sourcing local views also has considerable advantages: the use of local expert opinion increases the likelihood of successful bottom-up implementation and sustainability and it can help circumvent problems that top-down initiatives are often faced with [17, 37]. The fact that the research team already had an established working relation with many participants helped create an open atmosphere, but it could also have introduced facilitator and/or social desirability bias. The base CLDs were sent to group participants for validation along with a narrative, but the integrated CLD was not. As the latter was compiled by two of the authors, factors may have been unjustly matched. In the structural analysis we assumed that the plausibility of a path decreases with the length of that path (Formula 2). An alternative assumption could have been to use the minimum plausibility among its links instead. This would have made the impact of long-term effects larger. In the structural analysis we could not

investigate the relative strength of relationships or investigate time delays. Although the dynamic analysis did incorporate time delays, these were based on the research team's input rather than data or input from workshop participants. Many of the above limitations could be improved on by organizing more GMB workshops where the integrated CLD are evaluated by stakeholders, who then review and provide input to the dynamic analysis.

Future research should explore the expected long-term effects of the policy recommendations from this study. This would require data collection on key parameters, both through additional workshops and literature/database reviews. Here it would also be useful to consider external influences that may impact the surgical system, such as enrolment at teaching facilities and demographic trends. It would also be relevant to differentiate between types of DH situations (whereas in this study we looked more abstractly at an *average* Zambian DH). A second requirement would be an assessment of the different possible interventions along with their feasibility and costs. In our dynamic analysis, for example, only stimulating protocols had a quicker effect on volume than mentoring only. However, it is quite unlikely that policy can be developed that can impact *only* protocol adherence (and so strongly).

## Conclusions

To strengthen surgical capacity at the district level in support of a country's National Surgical Obstetric and Anesthesia Plan, a multi-pronged, multi-level surgical mentoring strategy is needed that takes the complex adaptive nature of the healthcare system into account. Coordination at the provincial level with accountability and support from the various groups of stakeholders–at national, provincial and district levels–is needed. Monitoring of important factors and periodic participatory evaluations of the surgical mentoring program are needed to foster a culture of continuous quality improvement.

## Supporting information

**S1 Data. All causal loop diagrams, code used in the analyses, and full results from quantitative analysis.**
(ZIP)

## Acknowledgments

We are grateful to the GMB participants for their valuable contributions. We thank Monic Lansu for the discussions about the design of the GMB workshops and Miranda Versteeg for her assistance with the visual layout of the integrated causal loop diagrams.

## Author Contributions

**Conceptualization:** Henk Broekhuizen, Martilord Ifeanyichi, Leon Bijlmakers.

**Data curation:** Henk Broekhuizen, Martilord Ifeanyichi, Mweene Cheelo, Grace Drury, Leon Bijlmakers.

**Formal analysis:** Henk Broekhuizen, Martilord Ifeanyichi.

**Funding acquisition:** John Kachimba, Ruairí Brugha, Jakub Gajewski, Leon Bijlmakers.

**Investigation:** Henk Broekhuizen, Martilord Ifeanyichi, Mweene Cheelo, Grace Drury, Leon Bijlmakers.

**Methodology:** Henk Broekhuizen, Etiënne Rouwette, Leon Bijlmakers.

**Project administration:** Mweene Cheelo.

**Resources:** Mweene Cheelo.

**Supervision:** Leon Bijlmakers.

**Writing – original draft:** Henk Broekhuizen, Martilord Ifeanyichi, Leon Bijlmakers.

**Writing – review & editing:** Henk Broekhuizen, Martilord Ifeanyichi, Mweene Cheelo, Grace Drury, Chiara Pittalis, Etiënne Rouwette, Michael Mbambiko, John Kachimba, Ruairí Brugha, Jakub Gajewski, Leon Bijlmakers.

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
