## [Decision Letter · Decision Letter 0]

28 Jun 2021

PONE-D-21-06371

Options for surgical mentoring: lessons from Zambia based on stakeholder consultation and systems science

PLOS ONE

Dear Dr. Broekhuizen,

Thank you for submitting your manuscript to PLOS ONE. After careful consideration, we feel that it has merit but does not fully meet PLOS ONE’s publication criteria as it currently stands. Therefore, we invite you to submit a revised version of the manuscript that addresses the points raised during the review process.

I have read with great interest the manuscript entitled ‘Options for surgical mentoring: lessons from Zambia based on stakeholder consultation and systems science’. In this original article, the authors apply a participatory approach from system dynamics, group model building workshops, to formulate policy recommendations on methods to coordinate and sustain surgical mentoring and monitor district-level surgical performance.

The manuscript is written well and the topic of clinical interest. The results reported are original and have not been published elsewhere. The study was conducted according to ethical standards.

The analyses and statistics were performed to a high technical standard and are described in sufficient detail.

The conclusions are supported by the data presented, and the authors acknowledge the limitations of the manuscript.

Reviewers raised some questions regarding previous work published by the group. Please, add further information in the Discussion about the novelty of this work and what was already previously published.

In addition, the expert reviewers asked for minor corrections in the manuscript, including further clarification on the Methods to ensure replicability of the study. Please carefully revise all comments.

To sum up, although concerns regarding novelty are not an exclusion criterion for publication, replication studies are not accepted. Therefore,  I would kindly ask the authors to reinforce the scientific rationale for the submitted work and clearly reference and discuss the existing literature.

Authors must address these points in the Discussion of the paper. Please provide a point-by-point response to reviewers’ comments.

We look forward to receiving your revised manuscript.

Kind regards,

Yuri Boteon, M.D., Ph.D.

Academic Editor

PLOS ONE

Journal Requirements:

Reviewers' comments:

Reviewer's Responses to Questions

**Comments to the Author**

1. Is the manuscript technically sound, and do the data support the conclusions?

Reviewer #1: Partly

Reviewer #2: Partly

Reviewer #3: Yes

2. Has the statistical analysis been performed appropriately and rigorously? 

Reviewer #1: Yes

Reviewer #2: Yes

Reviewer #3: Yes

3. Have the authors made all data underlying the findings in their manuscript fully available?

Reviewer #1: Yes

Reviewer #2: Yes

Reviewer #3: Yes

4. Is the manuscript presented in an intelligible fashion and written in standard English?

Reviewer #1: Yes

Reviewer #2: Yes

Reviewer #3: Yes

5. Review Comments to the Author

Reviewer #1: The manuscript has merit but unfortunately this area has been extensively researched within Zambia settings and also in other low middle income countries. There are many examples of recent published literature some which will be referenced below. The article unfortunately has very little to offer and adds nothing substantial to existing knowledge base. The manuscript can be considered for short communication. Examples of recent published literature similar to this area include:

Forcillo J, Watkins DA, Brooks A, Hugo-Hamman C, Chikoya L, Oketcho M, Thourani VH, Zühlke L, du Toit H, Nghaamwa J, Beshir S. Making cardiac surgery feasible in African countries: Experience from Namibia, Uganda, and Zambia. The Journal of thoracic and cardiovascular surgery. 2019 Nov 1;158(5):1384-93.

Gajewski J, Monzer N, Pittalis C, Bijlmakers L, Cheelo M, Kachimba J, Brugha R. Supervision as a tool for building surgical capacity of district hospitals: the case of Zambia. Human resources for health. 2020 Dec;18(1):1-8.

Gajewski J, Mweemba C, Cheelo M, McCauley T, Kachimba J, Borgstein E, Bijlmakers L, Brugha R. Non-physician clinicians in rural Africa: lessons from the Medical Licentiate programme in Zambia. Human resources for health. 2017 Dec;15(1):1-9.

Reviewer #2: 1. I suggest that the authors consider reviewing the title(see attached stick note in the manuscript).

2. Methods:

Kindly define how the participants in each group accepted a factor as influencing or were influenced by a central factor(line 146 to 155).Was it by consensus or majority votes approving the view point? A further clarification will help in replicability of the study

For the Methods deployed for the study, it may be difficult to avoid biases as it requires revision of participating group's close loop diagram by independent reviewers who may have inherent biases; it may be imperative to state the pedigree of the independent reviewers. This might help in the replicability of the study.

Reviewer #3: The article is acceptable as such, but it is long and complex, compared to what the reader can learn by reading it.

Only Error I found is that on the line 62 the word MARVEL is not preceded by words of which it is an abbreviation.

It is explained only at the line 217, "method to analyse relations between variables using enriched loops".

So this should be on line 62, with those words in Capital letters.

6. PLOS authors have the option to publish the peer review history of their article (what does this mean?). If published, this will include your full peer review and any attached files.

Reviewer #1: No

Reviewer #2: No

Reviewer #3: **Yes: **Mikko Kalevi Aalto

---

## [Author Response · Author response to Decision Letter 0]

2 Aug 2021

Dear editor,

Thank you for your handling of this manuscript and the opportunity to send in a revision. We also thank the reviewers for their time and insightful comments. Below we will address each of them in turn.

Kind regards also on behalf of my colleagues,

Dr. Henk Broekhuizen

Reviewer #1: The manuscript has merit but unfortunately this area has been extensively researched within Zambia settings and also in other low middle income countries. There are many examples of recent published literature some which will be referenced below. The article unfortunately has very little to offer and adds nothing substantial to existing knowledge base. The manuscript can be considered for short communication. Examples of recent published literature similar to this area include:

• Forcillo J, Watkins DA, Brooks A, Hugo-Hamman C, Chikoya L, Oketcho M, Thourani VH, Zühlke L, du Toit H, Nghaamwa J, Beshir S. Making cardiac surgery feasible in African countries: Experience from Namibia, Uganda, and Zambia. The Journal of thoracic and cardiovascular surgery. 2019 Nov 1;158(5):1384-93.

• Gajewski J, Monzer N, Pittalis C, Bijlmakers L, Cheelo M, Kachimba J, Brugha R. Supervision as a tool for building surgical capacity of district hospitals: the case of Zambia. Human resources for health. 2020 Dec;18(1):1-8.

• Gajewski J, Mweemba C, Cheelo M, McCauley T, Kachimba J, Borgstein E, Bijlmakers L, Brugha R. Non-physician clinicians in rural Africa: lessons from the Medical Licentiate programme in Zambia. Human resources for health. 2017 Dec;15(1):1-9.

Our response: We thank the reviewer for the comment. District-level surgery in sub-Saharan Africa has indeed received ample considerable and much needed attention from several scholars in the past couple of years, rightly so given the large unmet surgical need of rural dwellers. We are aware of the cited articles, two of which are (co-)authored by some of the authors of the present paper manuscript that is currently under review. We do not agree with the reviewer’s assertion that the current study adds little to the existing knowledge base, because there are several major differences as we will point out below. The present study adds also to the discussion around NSOAP implementation in Zambia, specifically on how the central level, combined with provincial-level authorities and surgical experts can support initiatives to increase access to surgical care for rural dwellers. We are also in the process of finalizing the first ever evaluation report pertaining to the implementation of the Zambian NSOAP. This paper provides new knowledge about obstacles and enablers to scale up surgery, with a particular focus on roles and insights that other stakeholders, who had not been invited to develop the Zambian NSOAP, could play in this process.

The proposed manuscript is also novel in the way it builds on the growing evidence of the importance of participatory action research (PAR) in scaling up surgical services. PAR in the form of GMB workshops enabled us to engage frontline providers and district level hospital managers in discussions about how best to address problems they reported themselves. This is not the dominant way of doing research in implementation sciences, and needs to be promoted and substantiated with evidence. This paper offers such an opportunity. 

The Gajewski et al. 2017 paper published in Human Resources for Health (HRH) was derived from the preceding COST-Africa intervention research project. This provided evidence, based on a one-off end-of-intervention set of qualitative interviews with district and national stakeholders, of the central role played in surgical service provision by medical licentiates (a type of non-physician clinician [NPC]) in district hospitals in Zambia. It also provided evidence about described some of the challenges faced by these cadres, beyond the typical shortages of infrastructure and supplies.

The need for ongoing surgical team mentoring, which was the main recommendation of the research, has proven to be a powerful tool to address some of these challenges. Insights from the COST-Africa project, which was originally designed as a training intervention (the context of the 2017 paper), were used to develop the mentoring intervention that lies at the heart of the SURG-Africa study, which in combination with the COST-Africa study provided the data for the 2020 HRH paper and which the present study investigates.

The Gajewski 2020 HRH paper, also from the COST-Africa project, investigated the experiences of clinical staff while they were receiving the supervision intervention in the Zambian setting. In some respects it covered the same ground as the HRH 2017 paper, drawing on the insights of individual district hospital-based NPCs, medical colleagues and supervisors. The project systematically collected evidence to support the proposition in the earlier paper. Human Resources for Health deemed that the 2020 paper added new knowledge to the 2017 paper published in the same journal.

The manuscript under review with PLOS ONE has a different objective, offers a different perspective, and utilises a different method(ology) than the two earlier papers published in HRH.

Firstly, the current study, rather than just reiterating the importance and value of supervision / mentoring of district surgical clinicians, set out to investigate and provide recommendations for the sustainable implementation of supervision intervention in the Zambian healthcare system. This was a specific knowledge gap and recommendation identified in the preceding 2020 HRH paper.

Secondly, while there was overlap in the cadres from which data were collected (district medical and NPC clinicians and specialist supervisors), there was a strong representation of middle and senior level managers and administrators in the GMB workshops, whose input and support would be essential to embedding and making such supervision sustainable. For those of us who have worked in health systems (research) in Africa for several decades, this is no easy ambition.

Thirdly, the 2020 (like the 2017) Gajewski et al study used a different method: individual semi-structured, qualitative interviews with clinical staff. In the current study we applied group model building (a novel method in surgical policy) with representatives from various clinical and administrative staff from several district and central hospitals, along with representatives from district councils and provincial administrations, each with their own mandates and in four different contexts. This enabled us to combine different perspectives while also taking into account the dynamic aspects of the Zambian surgical system with its local peculiarities to arrive at policy recommendations for sustainable implementation and replication of surgical mentoring.

The 2019 Forcillo et al. paper study is considerably different study from the manuscript currently under review with PLOS ONE in its focus, objectives and study method:

• Rather than focusing on supervision and mentoring of support to district-level hospitals for strengthening the provision of basic surgery, it focuses on extending cardiac surgery, either through outreach clinics conducted by specialists who are based at tertiary referral facilities, or through privately managed “health centres”.

• It lists the requirements (one of which is the establishment of training and mentorship programmes) for expanding specialist surgery, while our study takes a systems perspective on sustaining the implementation of a specific mentoring model that was trialed for two years in Zambia.

• Similar to our earlier 2017 and 2020 papers, published in Human Resources for Health, Forcillo et al. is based on in-depth interviews, along with facility surveys, whereas the present study utilises group model building to crack one of the outstanding challenges in health systems in Africa - i.e. embedding sustainable supervision systems for district clinical care in a national strategy.

Lastly, we would like to point out that another manuscript from our SURG-Africa team has recently been accepted for publication (by IJHPM) and it will soon be available online: 

• Broekhuizen H, Ifeanyichi M, Mwapasa G, Pittalis C, Noah P, Mkandawire N, Borgstein E, Brugha R, Gajewski J, Bijlmakers L. Improving access to surgery through surgical team mentoring – policy lessons from group model building with local stakeholders in Malawi.

Similar to the Zambia manuscript currently under review, this article uses group model building with a variety of local stakeholders as the main methodology. But it does this in a different context (Malawi) and with a different objective, i.e., to identify the most suitable scenario for a continuation and possible further expansion of surgical team mentoring in Malawi, as well as the future resource requirements of different scenarios and their ‘soft’ consequences. We do think that this article further adds to the slowly expanding evidence base.

Reviewer #2: 1. I suggest that the authors consider reviewing the title(see attached stick note in the manuscript). [I wish to suggest Policy options for surgical mentoring

My suggestion was based on the fact that the study aimed to explore policy recommendations which may impart on existing mentoring structure rather than identifying various options or alternate options for surgical mentoring]

Our response: We agree with the reviewer that this better fits the manuscript’s contents, we have changed the title.

2. Methods: Kindly define how the participants in each group accepted a factor as influencing or were influenced by a central factor(line 146 to 155).Was it by consensus or majority votes approving the view point? A further clarification will help in replicability of the study

Our response: The decision to include a factor in the causal loop diagram was taken by consensus. We have added this detail to the methods section (around line 161). 

For the Methods deployed for the study, it may be difficult to avoid biases as it requires revision of participating group's close loop diagram by independent reviewers who may have inherent biases; it may be imperative to state the pedigree of the independent reviewers. This might help in the replicability of the study.

Our response: we agree with this point, we have added this to the methods at line 138.

Reviewer #3: The article is acceptable as such, but it is long and complex, compared to what the reader can learn by reading it.

Only Error I found is that on the line 62 the word MARVEL is not preceded by words of which it is an abbreviation.

It is explained only at the line 217, "method to analyse relations between variables using enriched loops".

So this should be on line 62, with those words in Capital letters.

Our response: Thank you for your positive comments and suggestion. We have changed this on line 62.

---

## [Decision Letter · Decision Letter 1]

6 Sep 2021

Policy options for surgical mentoring: lessons from Zambia based on stakeholder consultation and systems science

PONE-D-21-06371R1

Dear Dr. Broekhuizen,

We’re pleased to inform you that your manuscript has been judged scientifically suitable for publication and will be formally accepted for publication once it meets all outstanding technical requirements.

Kind regards,

Yuri Longatto Boteon, M.D., Ph.D.

Academic Editor

PLOS ONE

Additional Editor Comments (optional):

Reviewers' comments:

Reviewer's Responses to Questions

**Comments to the Author**

1. If the authors have adequately addressed your comments raised in a previous round of review and you feel that this manuscript is now acceptable for publication, you may indicate that here to bypass the “Comments to the Author” section, enter your conflict of interest statement in the “Confidential to Editor” section, and submit your "Accept" recommendation.

Reviewer #1: All comments have been addressed

Reviewer #2: All comments have been addressed

Reviewer #3: All comments have been addressed

2. Is the manuscript technically sound, and do the data support the conclusions?

Reviewer #1: Yes

Reviewer #2: Yes

Reviewer #3: Yes

3. Has the statistical analysis been performed appropriately and rigorously? 

Reviewer #1: Yes

Reviewer #2: Yes

Reviewer #3: Yes

4. Have the authors made all data underlying the findings in their manuscript fully available?

Reviewer #1: Yes

Reviewer #2: Yes

Reviewer #3: Yes

5. Is the manuscript presented in an intelligible fashion and written in standard English?

Reviewer #1: Yes

Reviewer #2: Yes

Reviewer #3: Yes

6. Review Comments to the Author

Reviewer #1: Thank you for addressing the points highlighted in the previous detailed review. The manuscript is now suitable for publication in the journal.

Reviewer #2: (No Response)

Reviewer #3: I recommend this first revised version to be published in the present form. It is interesting and very relevant for those doing similar work, either informally or formally.

7. PLOS authors have the option to publish the peer review history of their article (what does this mean?). If published, this will include your full peer review and any attached files.

Reviewer #1: No

Reviewer #2: No

Reviewer #3: **Yes: **Mikko Kalevi Aalto

---

## [Editor Report · Acceptance letter]

20 Sep 2021

PONE-D-21-06371R1 

Policy options for surgical mentoring: lessons from Zambia based on stakeholder consultation and systems science 

Dear Dr. Broekhuizen:

I'm pleased to inform you that your manuscript has been deemed suitable for publication in PLOS ONE. Congratulations! Your manuscript is now with our production department. 

Kind regards, 

on behalf of

Prof. Yuri Longatto Boteon 

Academic Editor

PLOS ONE